# Self-Explaining Deviations for Coordination

**Hengyuan Hu**[*][†]
Stanford University
hengyuan@cs.stanford.edu

**Samuel Sokota**[*]
Carnegie Mellon University
ssokota@andrew.cmu.edu

**David Wu**
Meta AI
dwu@meta.com

**Anton Bakhtin**
Meta AI
yolo@meta.com

**Andrei Lupu**
Meta AI & FLAIR, University of Oxford
alupu@meta.com

**Brandon Cui**[†]
Mosaic ML
brandon@mosaicml.com

**Jakob N. Foerster**[†]
FLAIR, University of Oxford
jakob.foerster@eng.ox.ac.uk

## Abstract

Fully cooperative, partially observable multi-agent problems are ubiquitous in the real world. In this paper, we focus on a specific subclass of coordination problems in which humans are able to discover *self-explaining deviations* (SEDs). SEDs are actions that deviate from the common understanding of what reasonable behavior would be in normal circumstances. They are taken with the intention of causing another agent or other agents to realize, using theory of mind, that the circumstance must be abnormal. We motivate this idea with a real world example and formalize its definition. Next, we introduce an algorithm for improvement maximizing SEDs (IMPROVISED). Lastly, we evaluate IMPROVISED both in an illustrative toy setting and the popular benchmark setting Hanabi, where we show that it can produce so called *finesse* plays.

## 1 Introduction

Humans generally assume other humans follow certain social norms when acting in the society and interpret behavior using these norms to infer hidden information about the world. For example, when a driver sees a car ahead suddenly stopping on a road, they may infer that there is some accident ahead or the car is broken. However, humans also have the ability to improvise when the conventional actions are restricted or there exist other actions that can bring about superior outcomes. Taking the event in Figure 1 as an example, the caller deviates from the conventional practice of stating the situation truthfully because it would have negative consequences. This type of deviation involves two noteworthy aspects. First, it can be detected by the dispatcher because it appears to be a mistake under common practice. Second, the dispatcher's response can be independently decided by both individuals based on the common understanding of the world and the deviation chosen by the caller.

We refer to this type of phenomena as self-explaining deviations (SEDs). SEDs are actions that, under normal circumstances, would not make sense, given the agents' common understanding. They are executed with the intention that teammates will use *theory of mind* to deduce that the situation is unusual in a particular way, and will adapt their behavior to account for this additional information.

---

[*]Equal Contribution.
[†]Work done while at Meta AI.

36th Conference on Neural Information Processing Systems (NeurIPS 2022).

Dispatcher: Oregon 911.
Caller: I would like to order a pizza at...
Dispatcher: You called 911 to order a pizza?
Caller: Uh, Yeah, apartment...
Dispatcher: This is the wrong number to call for a pizza.
Caller: No no no... you're not understanding me.
Dispatcher: I'm getting you now. Is the other guy still there?
Caller: Yep. I need a large pizza.
Dispatcher: All right. How about medical. You need medical?
Caller: No. With pepperoni.
Dispatcher: Turn your sirens off before you get there. Caller ordered a pizza. And agreed with everything I said that there's domestic violence going on.

Figure 1: A real-life self-explaining deviation.

For our contribution, we first formalize the problem setting and definition of SEDs. Then we introduce a novel planning algorithm, IMPROVement maxImizing Self-Explaining Deviations (IMPROVISED), for performing SEDs. We show that, under some assumptions, IMPROVISED performs the optimal SED in terms of expected return maximization. Next, we provide a motivating experiment, illustrating that in a small toy problem designed to require SEDs to perform optimally, IMPROVISED is able to compute an optimal joint policy, whereas other multi-agent learning algorithms are not. Lastly, we present experiments on the large scale benchmark Hanabi [1], where we show that IMPROVISED is able to produce *finesse* plays, which is one of the most interesting techniques that human experts perform frequently.

## 2 Background

**FOSG and Public POMDP** For our notation, we use an adaption of factored observation stochastic games (**FOSG**) [7]. $\mathcal{W}$ is the set of **world states** and $w^0$ is a designated initial state. $\mathcal{A} = \mathcal{A}_1 \times \cdots \times \mathcal{A}_N$ is the space of **joint actions**. $\mathcal{T}$ is the **transition function** mapping $\mathcal{W} \times \mathcal{A} \to \Delta(\mathcal{W})$. $\mathcal{R} \colon \mathcal{W} \times \mathcal{A} \to \mathbb{R}$ is the **reward function**. $\mathcal{O} = (\mathcal{O}_{\mathrm{priv}(1)}, \ldots, \mathcal{O}_{\mathrm{priv}(N)}, \mathcal{O}_{\mathrm{pub}})$ is the **observation function** where $\mathcal{O}_{\mathrm{priv}(i)} \colon \mathcal{W} \times \mathcal{A} \times \mathcal{W} \to \mathbb{O}_{\mathrm{priv}(i)}$ specifies the **private observation** that player $i$ receives. $\mathcal{O}_{\mathrm{pub}} \colon \mathcal{W} \times \mathcal{A} \times \mathcal{W} \to \mathbb{O}_{\mathrm{pub}}$ specifies the **public observation** that all players receive. $O_i = \mathcal{O}_i(w, a, w') = (\mathcal{O}_{\mathrm{priv}(i)}(w, a, w'), \mathcal{O}_{\mathrm{pub}}(w, a, w'))$ is player $i$'s **observation** and a **history** is a finite sequence $h = (w^0, a^0, \ldots, w^t)$. The **set of histories** is denoted by $\mathcal{H}$. The **information state** for player $i$ at $h = (w^0, a^0, \ldots, w^t)$ is $s_i(h) \coloneqq (O_i^0, a_i^0, \ldots, O_i^t)$. The **information state space** for player $i$ is $\mathcal{S}_i \coloneqq \{s_i(h) \mid h \in \mathcal{H}\}$. The **legal actions** for player $i$ at $s_i$ is denoted $\mathcal{A}_i(s_i)$. A **joint policy** is a tuple $\pi = (\pi_1, \ldots, \pi_N)$, where **policy** $\pi_i$ maps $\mathcal{S}_i \to \Delta(\mathcal{A}_i)$. The **public state** at $h$ is the sequence $s_{\mathrm{pub}}(h) \coloneqq s_{\mathrm{pub}}(s_i(h)) \coloneqq (O_{\mathrm{pub}}^0, \ldots, O_{\mathrm{pub}}^t)$. The **information state set** for player $i$ at $s \in \mathcal{S}_{\mathrm{pub}}$ is $\mathcal{S}_i(s) \coloneqq \{s_i \in \mathcal{S}_i \mid s_{\mathrm{pub}}(s_i) = s\}$, where $\mathcal{S}_{\mathrm{pub}}$ is the space of public states. Finally, the **reach probability** of $h$ under $\pi$ is $P^\pi(h)$.

Rather than working with in a multi-agent setting directly, we invoke the public POMDP transformation that maps cooperative multi-agent settings to equivalent single-agent POMDPs. Given a common-payoff FOSG $\langle \mathcal{N}, \mathcal{W}, w^0, \mathcal{A}, \mathcal{T}, \mathcal{R}, \mathcal{O} \rangle$, we can construct an equivalent **public POMDP** [10] $\langle \tilde{\mathcal{W}}, \tilde{w}^0, \tilde{\mathcal{A}}, \tilde{\mathcal{T}}, \tilde{\mathcal{R}}, \tilde{\mathcal{O}} \rangle$ as follows: The world states of the public POMDP $\tilde{\mathcal{W}}$ is the set $\{(s_1(h), \ldots, s_N(h)) : h \in \mathcal{H}\}$. The initial world state of the public POMDP $\tilde{w}^0$ is the tuple $(s_1(h^0), \ldots, s_N(h^0))$.

The actions of the public POMDP are called **joint prescriptions**. It is denoted by $\Gamma$ and has $N$ components. The $i$th component of it $\Gamma_i$ is the **prescription** for player $i$. A prescription $\Gamma_i$ maps $s_i$ to an element of $\mathcal{A}_i(s_i)$ for each $s_i \in \mathcal{S}_i(s_{\mathrm{pub}}(h))$; it instructs a player in the common-payoff FOSG how to act as a function of its private information. Given $\tilde{w} \equiv (s_1, \ldots, s_n)$ and $\Gamma$, the transition distribution $\tilde{\mathcal{T}}(\tilde{w}, \Gamma)$ is induced by $\mathcal{T}((s_1, \ldots, s_n), a)$, where $a \equiv \Gamma((s_1, \ldots, s_n)) \coloneqq (\Gamma_1(s_1), \ldots, \Gamma_N(s_N))$[3]. Given $\tilde{w} \equiv (s_1, \ldots, s_n)$ and $\tilde{w}' \equiv (s_1', \ldots, s_n')$, the reward and observa-

---

[3] If original game was sequential, prescriptions for non-acting players map to "no-ops".

tion are given by $\tilde{\mathcal{R}}(\tilde{w}, \Gamma, \tilde{w}') \equiv \mathcal{R}((s_1, \ldots, s_n), \Gamma((s_1, \ldots, s_n)), (s'_1, \ldots, s'_n))$ and $\tilde{\mathcal{O}}(\tilde{w}, \Gamma, \tilde{w}') \equiv \mathcal{O}_{\text{pub}}((s_1, \ldots, s_n), \Gamma((s_1, \ldots, s_n)), (s'_1, \ldots, s'_n))$, respectively.

Every policy in the public POMDP corresponds to a joint policy in the underlying common-payoff FOSG, which receives exactly the same expected return. Therefore, it is sufficient to work with the public POMDP. See [14, 15] for further discussion.

When the public POMDP is considered as a belief MDP, rather than as a POMDP, its belief states are of the form $b^t \colon (s_1^t, \ldots, s_n^t) \mapsto P^{\Gamma^0, \ldots, \Gamma^{t-1}}(s_1^t, \ldots, s_N^t \mid s_{\text{pub}}^t)$. In words, this is the joint distribution over private information states, conditioned on the historical policy $(\Gamma^0, \ldots, \Gamma^{t-1})$ and the public state $s_{\text{pub}}^t$. This public belief MDP is abbreviated as the PuB-MDP.

## 3 Self-Explaining Deviations

While the name *self-explaining deviation* (SED) is novel to this work, the idea behind SEDs is not. One example of SEDs comes from the cooperative card game Hanabi [1] in the form of a play called a "finesse". Readers unfamiliar with Hanabi and finesse may first jump to Section 5.2 and Section 5.3 for the detailed descriptions. In a finesse play, the acting player, i.e., the first player who initiates the deviation as part of the finesse, intentionally communicates misleading information to a receiving player, which would hurt the team's score if the receiving player acted upon it using the established convention. The second player in the finesse, who acts after the first player but before the receiving player, realizes that the first player must have deviated from the established convention after observing the seemingly disastrous move. Knowing that the first player is rational, the second player realizes that there must be a way that they can also deviate to reach a better outcome than the one that the original convention would have led to. After the second player plays its part of the joint deviation, the original information from the first player is no longer misleading and the subsequent players can continue to follow the prior conventions.

SEDs may take place in any cooperative situation with at least two players where common knowledge blueprint policies that players follow under normal assumptions exist. Intuitively, SEDs capture a form of joint deviations where one agent takes an action that at first appears to be a mistake or otherwise highly off-policy from another player's perspective, i.e. zero or low probability under the blueprint. On the presumption that the first agent chose that action intelligently and deliberately and there is an off-policy action for the observing agent that could potentially result in an even-better-than-normal outcome for both, the observing agent may reason that the first agent "intends" them to take it—to take a leap of faith that the first agent has not erred, but rather knows both agents can get that better outcome, even if the observing agent does not have the information themselves to prove that this outcome will result. This work uses the term SEDs to describe this phenomenon of communication via apparent mistakes in a general context. We formalize SED in Section 3.1 and propose IMPROVISED, a novel planning algorithm that performs the optimal SEDs under some assumptions, in Section 4.

### 3.1 Examining Self-Explaining Deviations

We are now ready to investigate SEDs. To facilitate our investigation, let's consider the *trampoline-tiger* game in Figure 2 as an example. In this game, Alice is standing on a balcony, while Bob stands on the ground next to a lever. Pulling the lever will either deploy a trampoline below the balcony, or release a tiger. From above, Alice can observe which is the case, and then Alice decides whether to jump off the balcony (Y) or not (N). Bob *observes Alice's choice* but does *not* know whether the lever will deploy a trampoline or tiger, and decides to pull the lever (Y) or not (N). Alice wants to get down from the balcony but will die from the fall unless Bob pulls the lever *and* it releases a trampoline. If Bob pulls the lever and releases a tiger, it will eat Bob.

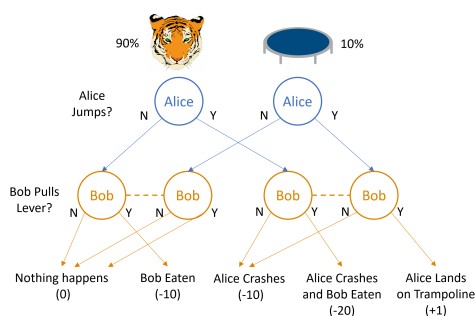

Figure 2: Alice knows whether Bob's lever will deploy a tiger or a trampoline, and then jumps or not. Bob observes only Alice's action and then pulls the lever or not. Dotted lines between two Bob nodes means that Bob cannot distinguish them.

Consider the joint policy $(* \mapsto \mathrm{N}, * \mapsto \mathrm{N})$. While not optimal, this joint policy is not unreasonable, as Bob won't be eaten and Alice won't fall to her death. Indeed, independent of the probability of a trampoline, this joint policy is a Nash equilibrium. In addition, as we shall see in Section 5.1, a range of MARL algorithms converges to this solution. However, there is a clear opportunity for a SED under this blueprint. If there is a tiger, Alice should obviously never jump. So if Bob observes Alice choosing to jump and trusts that Alice is intelligent and would only have chosen to jump if it could lead to a better result, Bob should realize that there must be a trampoline. Therefore Alice should trust that Bob will pull the lever if she jumps, as a result.

Under any planning algorithm that considers only unilateral deviations, Bob and Alice can't escape the original local optimum. Alice will never jump because she is under the belief that Bob will choose not to pull the lever.Therefore, finding SEDs in general requires considering multi-lateral deviations, i.e. simultaneous deviations by more than one player.

Since finding multi-lateral deviations in general games with partially observable state is a complex problem, we restrict our focus to settings satisfying the following assumptions:

**Assumption 1.** *We assume sequential and publicly observable actions.*

SEDs may still be possible with partially-observable actions, but for simplicity we focus on when Bob can directly observe Alice deviate from the blueprint, rather than having to deduce it from partial or incomplete observations.

**Assumption 2.** *We consider only cases where Bob's private information is not necessary for coordinating on the SED.*

For simplicity, we always refer to the first player who initiates the joint deviation as the *Alice* and the second player who figures out Alice's intention and plays their part of the joint deviation as *Bob* in the following discussions beyond the scope of this toy game. In other words, we only consider SEDs where Bob needs only act as a function of the common-knowledge state and the Alice's action, rather as a function of private knowledge of their own.

Under these assumptions, we define the SED as follows:

**Definition 1.** *Given a common knowledge blueprint (BP) policy $\pi$, common public belief of Alice and Bob $b$, and information state of Alice $s_1$, SED is any pair of joint deviations $(a_1', a_2')$ for Alice and Bob that satisfy all the following conditions:*

- *$q_\pi(b, s_1, a_1', a_2') \geq q_\pi(b, s_1)$, i.e., it gives higher expected future return than BP;*
- *$P(a_1' \mid \pi, b) \leq \epsilon$, i.e., $a_1'$ is highly unlikely under $\pi$ under the common belief;*
- *$a_2' \sim f(b, a_1')$, i.e., $a_2'$ is a function of the public belief and Alice's deviation action.*

Here, $q_\pi(b, s_1, [a_1, a_2])$ is Alice's estimate of the expected future return after executing the optional $a_1, a_2$ and following $\pi$ afterwards, and $\epsilon$ is a hyper-parameter. The function $f$ is determined by the algorithm designer under the constraint that $f$ only takes $b$ and $a_1'$ as input. It is used both by Alice to predict Bob's response and by Bob to decide the response independently.

## 4 IMPROVISED

In this section we derive IMPROVISED, an algorithm to perform the SEDs. The core of the algorithm is to find a function $f$ under the constraint of SED and an optimization procedure that Alice uses to decide whether to deviate given her information state $s_1$ and possible responses $a_2 \sim f$ from Bob.

### 4.1 Defining the Optimization Problem

With the observations in the prior sections, we are ready to write selecting a SED as a PuB-MDP optimization problem:

$$
\begin{aligned}
& \max_{\Gamma_1, \Gamma_2} \max[q_\pi(b, \Gamma_1, \Gamma_2), q_\pi(b)] \\
= & \max_{\Gamma_1, \Gamma_2} \mathbb{E}_{s_1 \sim b} \max\left[q_\pi(b, s_1, \Gamma_1(s_1), \Gamma_2), q_\pi(b, s_1)\right] \\
= & \max_{\Gamma_1, \Gamma_2} \mathbb{E}_{s_1 \sim b} \max\left[q_\pi(b, s_1, \Gamma_1(s_1), \Gamma_2 \circ \Gamma_1(s_1)), q_\pi(b, s_1)\right].
\end{aligned} \tag{1}
$$

where $\Gamma_1$ is the prescription for the acting player at current time step and $\Gamma_2$ is the prescription for the acting player at the subsequent time step, $\circ$ denotes function composition, and, in accordance with Definition 1, $\Gamma_2$ ranges over the space of $\mathcal{A}_1$. The series of equalities above begins with a generic multi-lateral search in the PuB-MDP. The LHS is ranging over Alice and Bob's prescriptions such that, if Alice uses $\Gamma_1$ and Bob uses $\Gamma_2$ in belief state $b$, and all the players play according to $\pi$ thereafter, the expected return is maximized. The first equality holds simply based on the fact that Alice is able to observe $s_1$ and the fact Alice can always decide to opt into the blueprint (because Bob knows which actions are supported by the blueprint, he can opt in exactly when Alice opts in, and play according to the deviation otherwise). The second equality holds by applying Assumptions 1 and 2, which imply both that Bob can observe Alice's action and that it is the only thing he needs to condition his decision upon (note that the change of the set over which $\Gamma_2$ ranges is left implicit).

## 4.2 An Easier Special Case

Unfortunately, expression (1) remains difficult to optimize, as the number of $\Gamma_1$ and $\Gamma_2$ is combinatorial. To ameliorate, we can apply the simplifying constraint of only allowing Alice and Bob to deviate for a single action pair regardless of Alice's information state $s_1$. For simplicity, we also set $\epsilon = 0$. Let $\mathcal{D}(b)$ be the set of plausible deviation actions for Alice, i.e., action that cannot be played under blueprint: $\mathcal{D}(b) := \{a_1 \in \mathcal{A}_1 | \forall s_1 \in b : \pi(a_1|s_1) = 0\}$. Let $\mathcal{R}$ be the set of all plausible actions that Bob can play in response in every possible state, i.e., $\mathcal{R} := \bigcap_{s_2 \in \mathcal{S}_2} \mathcal{A}_2(s_2)$. If $\mathcal{R} = \emptyset$, Alice will skip searching for SED this turn and follow BP. Now we can augment the expected value of a deviation with the allowed range of actions:

$$\hat{q}_\pi(b, s_1, a_1, a_2) = \begin{cases} q_\pi(b, s_1, a_1, a_2), & \text{if } a_1 \in \mathcal{D}(b), a_2 \in \mathcal{R}, \\ -\infty, & \text{otherwise.} \end{cases} \tag{2}$$

Then we can rewrite equation (1) using our newly defined $q$ function as follows:

$$(a_1^*, a_2^*) = \operatorname{argmax}_{a_1, a_2} \mathbb{E}_{s_1 \sim b} \max[\hat{q}_\pi(b, s_1, a_1, a_2), q_\pi(b, s_1)]. \tag{3}$$

These values could be found in $O(|\mathcal{S}_1||\mathcal{A}_1||\mathcal{A}_2|)$ time, given the two inner $q$ functions. Moreover, as all of these depend only on the public belief state, both players can compute them independently, assuming that there are no ties.

Once the deviation pair is computed, Alice decides whether to proceed with the deviation or not given her private information: Alice plays $a_1^*$ if $\hat{q}_\pi(b, s_1, a_1^*, a_2^*) > q_\pi(b, s_1)$ and the blueprint action otherwise. It is possible that $(a_1^*, a_2^*)$ is not a valid deviation pair if no plausible deviations exist, e.g., $a_1^*$ could be outside of $\mathcal{D}(b)$. However, in this case $\hat{q}_\pi(b, s_1, a_1^*, a_2^*)$ is $-\infty$, and so Alice will resort to playing blueprint. Bob can detect the deviation as $\mathcal{D}(b)$ is public knowledge and $a_1^* \in \mathcal{D}(b)$. Therefore, depending on his observation, he can either play $a_2^*$ or respond as usual.

We can represent the value of the policy that allows a single deviation at $b$ and plays blueprint afterwards as: $q_\pi^*(b) = \max_{a_1, a_2} \mathbb{E}_{s_1 \sim b} \max[\hat{q}_\pi(b, s_1, a_1, a_2), q_\pi(b, s_1)]$. It is easy to see that $q_\pi^* \geq q_\pi$, i.e., this algorithm has a weak policy improvement guarantee.

If we apply this procedure to the tiger-trampoline game with a blueprint policy that always no-ops:

$$\begin{aligned} q_\pi^*(b) &= \max_{a_1, a_2} \mathbb{E}_{s_1 \sim b} \max[\hat{q}_\pi(b, s_1, a_1, a_2), q_\pi(b, s_1)] \\ &= \max_{a_1, a_2} [P(\text{tiger}) \max(\hat{q}_\pi(b, \text{tiger}, a_1, a_2), 0) \\ &\qquad + P(\text{trampoline}) \max(\hat{q}_\pi(b, \text{trampoline}, a_1, a_2), 0)] \\ &= P(\text{trampoline}). \end{aligned}$$

As expected, we obtain a policy improvement with value equal to the probability of a trampoline. We discover the SED where Alice jumps when a trampoline is present and Bob pulls the lever. We provide a proof-of-principle implementation of IMPROVISED in the trampoline-tiger game that can be run online at `https://bit.ly/3KtMLT6`.

## 4.3 Coordination by Extending Conventions

In a coordination context, we cannot require Bob to perform arbitrary symmetry breaking, i.e., choose an action among several with the same expected value. For example, consider a case in which a

set of multiple action pairs tied for having the maximal value $\{(a_1 = x, a_2 = y), (a_1 = x, a_2 = z), (a_1 = w, a_2 = z)\}$. Say that Alice picks $a_1 = x$. This is problematic for Bob because he has no information about whether Alice is selecting $x$ because she is in an information state in which $(x, y)$ is good or whether she is in an information state in which $(x, z)$ is good (the states for which they are good may be disjoint).

One way to resolve this issue is to force Alice to respect Bob's inability to break ties by making the response function stochastic, e.g., a $\mathrm{softmax}$ over the expected values. That is, given that Alice deviates with $a_1$, Alice assumes that Bob plays

$$a_2^* \sim \mathrm{softmax}_{a_2} \left[ \mathbb{E}_{s_1 \sim b} \max[\hat{q}_\pi(b, s_1, a_1, a_2), q_\pi(b, s_1)]/t \right] \tag{4}$$

with a temperature hyper-parameter $t$ to control the sharpness of the distribution. Then Alice chooses to deviate using

$$\mathrm{argmax}_{a_1} \mathbb{E}_{s_1 \sim b} \max[\mathbb{E}_{a_2 \sim a_2^*(b, a_1)} \hat{q}_\pi(b, s_1, a_1, a_2), q_\pi(b, s_1)]. \tag{5}$$

Bob can either play according to the softmax or select the single maximum, as Alice has already picked the deviation assuming that Bob cannot break symmetries.

### 4.4 Taking Alice's Information State Into Account

Our discussion above has largely ignored the fact that Alice can perform different deviations given her information state $s_1$ (i.e., private observation). Define the response function $f(b, a_1) = \mathrm{softmax}_{a_2} \left[ \mathbb{E}_{s_1 \sim b} \max[\hat{q}_\pi(b, s_1, a_1, a_2), q_\pi(b, s_1)]/t \right]$. Given that Alice knows Bob will respond according to $a_2 \sim f(b, a_1)$, Alice is now free to optimize her action by taking her information state back into consideration. By applying Jensen's inequality, we observe that

$$\max_{a_1} \mathbb{E}_{s_1 \sim b} \max \left[ \mathbb{E}_{a_2 \sim f(b, a_1)} \hat{q}_\pi(b, s_1, a_1, a_2), q_\pi(b, s_1) \right]$$
$$\leq \mathbb{E}_{s_1 \sim b} \max_{a_1} \max \left[ \mathbb{E}_{a_2 \sim f(b, a_1)} \hat{q}_\pi(b, s_1, a_1, a_2), q_\pi(b, s_1) \right].$$

Therefore, at information state $s_1$, Alice opts out of the blueprint if the *optimal deviation action*, $a_1^* = \max_{a_1} \mathbb{E}_{a_2 \sim f(b, a_1)} \hat{q}_\pi(b, s_1, a_1, a_2)$, exceeds the expected return of the BP, $q_\pi(b, s_1)$. This is the final formulation of IMPROVISED$^E$. Please refer to the Appendix A for the detailed pseudocode.

We also define IMPROVISED$^P$, where Bob plays according to the probability of improvement

$$a_2^* \sim \mathrm{softmax}_{a_2} \left[ \mathbb{E}_{s_1 \sim b} I[\hat{q}_\pi(b, s_1, a_1, a_2) > q_\pi(b, s_1)]/t \right] \tag{6}$$

where $I$ is the indicator function and Alice selects any action maximizing the expected improvement. While IMPROVISED$^P$ does not maximize the expected return as IMPROVISED$^E$ does, it may better reflect human SEDs since it finds deviation pairs that maximize the *probability of improvement*.

## 5 Experiments

We test the IMPROVISED in two different settings. The first setting is the trampoline-tiger game explained before. Secondly, we apply IMPROVISED to three-player Hanabi, where we start from a blueprint trained on human data. We provide the code for our Hanabi experiments at https://github.com/facebookresearch/off-belief-learning/blob/main/pyhanabi/finesse.py.

### 5.1 Trampoline Tiger

As illustrated in Section 4.2, in the trampoline-tiger game IMPROVISED$^E$ and IMPROVISED$^P$ both recover the optimal SED, when starting from the no-op BP in $100\%$ of the runs. Alice will decide to jump whenever there is a trampoline and Bob then pulls the lever to open the door, leading to the optimal expected return of 0.1.

For comparison, we also ran 20 unique seeds of MAPPO [18] and 24 different hyper-parameter combinations of QMIX [12] on the trampoline-tiger problem. The results are shown in Figure 3. All runs converged rapidly to a policy that avoids the highly negative payoffs, but only 2/20 of the

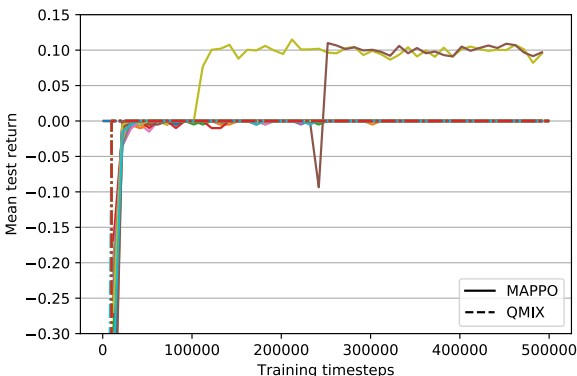

Figure 3: The plot shows 20 independent runs of MAPPO (solid lines) and 24 of QMIX (dashed lines) on the tiger-trampoline toy problem. Only 2 MAPPO runs find the optimal strategy while no QMIX run does.

MAPPO runs and 0/24 of the QMIX runs discovered the optimal strategy within 500k timesteps, leading to an average reward across runs of 0.01 for MAPPO and 0.0 for QMIX.

In our experiments, a variety of other standard multi-agent learning techniques could not solve this toy problem at all, including value-decomposition network [17] and simplified action decoder [4].

## 5.2 Hanabi

Hanabi is a benchmark challenge for multi-agent coordination. Briefly, in Hanabi, there is a 50 card deck consisting of 5 different suits (colors) and 5 ranks of cards within each suit, for a total of 25 unique combinations, with some duplicates. Two to five players cooperatively take turns playing cards, giving hints to other players, or discarding. Players may see their partners' hands, but *not* their own. The team's goal is to play exactly one card of each rank in each suit in increasing order, scoring 1 point per successful play. Upon three plays of cards duplicated or out-of-order, the players lose and instead score 0. Giving a hint spends one of 8 shared hint tokens, allowing one to name a color or rank in a partner's hand and indicate all cards of that color or rank in that hand. Discarding a card replenishes one hint token. For a more thorough description of the rules and basic strategy see [1].

## 5.3 What is a Finesse?

As mentioned earlier in Section 3, a clear example of a SED is the finesse move in Hanabi. We will recap it briefly here, for a more detailed explanation, see [1]. For convenience, we always refer to the acting player of a given turn as *player 1* and the players who move subsequently as *player 2*, *player 3*, and so forth.

The standard, vanilla form of the finesse in Hanabi occurs when *player 1* gives a hint to *player 3* (i.e. the player who moves two steps later) that, under the conventions common between the players, implies that *player 3* should play a particular card. Unbeknownst to *player 3*, that card, if played, would be out-of-order and would fail. Instead, *player 2* has just drawn the card that is both playable and if played makes *player 3*'s card playable next. *Player 2* is expected to realize that the only way that *player 1*'s hint could be good rather than a failure is if *player 2* themself has newly drawn this exact card, and to take a leap of faith and play that newly drawn card blindly, having no explicit information about it. The end result is that *player 1* signals in only one action for both other players to each successfully play a card—a large gain.

While other finesse-like patterns are possible, in our Hanabi experiments for simplicity we focus only on those finesse moves that follow the pattern described above.

| Model | Finesse-able | Finesse-complete |
|---|---|---|
| SAD [4] | 1375 | 597 (43.42%) |
| Other-Play [5] | 1537 | 512 (33.31%) |
| OBL (Level 5) [6] | 1100 | 356 (32.36%) |
| Behavior Clone [6] | 1376 | 1183 (85.97%) |

| Method | Finesse |
|---|---|
| Blueprint | 0 |
| SPARTA [8] | 52 |
| IMPROVISED$^E$ | 264 |
| IMPROVISED$^P$ | 269 |

(a) Number of finesse-able situations over 1000 games and % of those where *player 3* will correctly respond to complete the finesse if *player 1* and *2* were forced to play the finesse move. The Behavior Clone model has a much higher chance to complete a finesse.

(b) Number of finesses executed by different methods at 1000 finesse-complete situations using Behavior Clone agent from Table 1a as blueprint.

Table 1: IMPROVISED for Finesse in Hanabi

## 5.4 IMPROVISED in Hanabi

We design experiments to evaluate IMPROVISED in Hanabi from both qualitative and quantitative perspectives. In the qualitative evaluation, we study whether IMPROVISED can indeed perform finesses in manually selected situations where finesse could be completed (finesse-complete situations). We are particularly interested in the finesse style SEDs as they are the most natural and intuitive way to demonstrate IMPROVISED's ability to perform SEDs. Note that IMPROVISED discovers many types of SEDs in Hanabi beyond the finesse style moves because it searches for any beneficial joint deviations that when *player 1* initiates a deviation, *player 2* can independently figure out the correct deviation in response. However, other types of SEDs may be hard for humans to interpret and less intuitive to analyze and present. In the quantitative evaluation, we show that IMPROVISED as a planning algorithm improves the expected return when applied to test finesse-able situations and entire games.

To implement IMPROVISED in Hanabi, we first need a belief function from which we can sample game states given either public or private knowledge of the game to perform Monte Carlo rollouts. Luckily, the belief over possible hands in Hanabi can be computed analytically [8]. The belief is first initialized to cover all possible hands and then incrementally updated by filtering out hands that contradict with public knowledge revealed through hints and that would have caused the players to pick different moves at each time step. Since we are mainly concerned with the finesse style SEDs in the experiments, we restrict the search action space $\mathcal{A}_1$ to contain only the hint moves that target *player 3* and $\mathcal{A}_2$ to contain only the play moves. *Player 1* and *player 2* compute their own copies of the response function $f$ independently. The detailed hyper-parameters and computational cost are in Section C.

**Finessable Situation Experiments** To check IMPROVISED's ability to perform finesse, we first generate situations where seasoned human players may carry out finesse moves. We use a blueprint policy $\pi$ to generate selfplay games over a range of decks (game seeds) and look for situations where *player 1* observes that *player 2*'s newest draw card is playable and has not been hinted at and where *player 3* holds an un-hinted card that can be played after *player 2* plays their newest card (finesse-able situations). Then we manually override *player 1*'s and *player 2*'s moves in finesse-able situations to carry out the finesse and check if *player 3* will play the designated card under the blueprint to make the finesse complete and worthwhile (finesse-complete situations). We experiment with 4 different blueprints and show their statistics of finesse-complete situations over 1000 game seeds in Table 1a. All the agents are trained following the settings in corresponding prior works. Although finesse-able situations are common in the games played by all four agents, the first three RL agents trained without human data rarely complete a manually enforced finesse move, making them undesirable for the subsequent experiment. It is also worth noting that none of the RL blueprints perform finesses by themselves while the behavior clone agent performs only 1 finesse out of the 1183 finesse-complete situations.

**Finesse Execution Experiments** We apply both IMPROVISED$^E$ and IMPROVISED$^P$ with the behavior clone blueprint on 1000 randomly chosen finesse-complete situations to check how many finesse it performs. The results are shown in Table 1b. Due to the novelty of this problem setting, there has been no prior method to directly compare against. For reference, we run SPARTA [8], a strong search algorithm for Dec-POMDP designed to find unilateral deviations that maximize the expected return, on the same situations with the same restrictions on the set of actions that the agents may deviate to. From the table, we see that both IMPROVISED algorithms perform significantly more finesses, indicating their effectiveness in finding SEDs on the fly. In the situations where

IMPROVISED does not perform finesses, it can either be that IMPROVISED finds no beneficial deviations or it finds better, non-finesse SEDs. It is also interesting to see that IMPROVISED$^P$ finds roughly the same amount of finesses as IMPROVISED$^E$.

**Full Game Experiments** Then, we run IMPROVISED on the full game of Hanabi. At any time step, the active player will first check whether a deviation has been initiated by other players by checking whether previous players have picked actions with low probability under the blueprint. If a deviation has been initiated, they will play their corresponding role as either *player 2* or *player 3*. Otherwise the active player will decide whether to deviate using the IMPROVISED method. Over a fixed set of 100 game seeds, IMPROVISED$^E$ and IMPROVISED$^P$ perform **29** and **21** finesses respectively, while SPARTA only performs 3. Although it is still much less frequent than what expert human players will do, IMPROVISED is nonetheless a meaningful step along this novel and challenging direction.

Although our work is focused on understanding and replicating this human capability rather than optimizing for self-play score, we also report quantitative results showing a noticeable increase in score using IMPROVISED compared to the plain blueprint. If we run IMPROVISED only at the 1000 finesse-complete situations from Table 1b and use blueprint for the rest of the timesteps, then IMPROVISED$^E$ and IMPROVISED$^P$ achieve average scores of $18.08 \pm 0.28$ and $18.18 \pm 0.27$ respectively while the blueprint gets $17.18 \pm 0.28$ on those same games. When we run IMPROVISED$^E$ on the full game, we improve the average score from $17.80 \pm 0.85$ to $23.54 \pm 0.14$.

# 6    Related Work

**Public Belief Methods** Our method builds on prior work that models Dec-POMDPs as PuB-MDPs [10, 11, 2, 3, 15, 16]. PuB-MDPs enable theoretically sound planning algorithms without having to reason about the agents' entire policies because the public belief state serves as a sufficient statistic for planning. For example, SPARTA [8] conducts a one-step lookahead starting from the public belief state and chooses the action that maximizes expected value assuming all players play according to a common-knowledge BP thereafter. However, SPARTA searches only for unilateral deviations from the BP rather than multilateral deviations, because searching over all of the latter would be intractable. As discussed in Section 3.1, discovering SEDs may require considering multi-lateral deviations from the BP. IMPROVISED is able to discover these multi-lateral deviations by searching over a constrained set of multi-lateral deviations.

**Human-Like Coordination** Outside of this work, another notable work in the direction of incorporating human-like behavior into AI agents is that of Ma et al. [9]. Ma et al. [9] show that certain architectures have better inductive biases for respecting the correspondence between action features and observational features. Our work is complementary in the sense that we investigate a phenomenon that occurs in unusual situations, whereas Ma et al. [9] investigate a phenomenon that occurs by default.

# 7    Conclusions

Coordinating with others using minimal explicit agreement and extending conventions "on the fly" is one of the most intriguing reasoning capabilities. In this paper we formalize the definition of such behavior as *self-explaining deviations*. We showed that existing methods tend not to perform these types of deviations and presented an algorithm called IMPROVISED that can both discover them and respond correctly when they are being carried out by other agents at test time.

**Limitations and Future Work** We see two main limitations of IMPROVISED that would be worth addressing in future work. The first is the fact that IMPROVISED requires exact knowledge of all teammates' policies and assumes that other teammates are also running IMPROVISED. In real coordination settings, it is both unrealistic to assume exact knowledge of the policies of externally specified teammates and unrealistic to assume that those teammates are implementing the same SED algorithm. The second is the computational cost of IMPROVISED, as it requires Monte Carlo rollouts to estimate many crucial quantities at test time (more details in the appendix). One concrete direction toward addressing this issue is to derive a learning based version of IMPROVISED by learning the Q-function $q(b, s_1, a_1, a_2)$.

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
