# A  Pseudocode of IMPROVISED$^E$

---

**Algorithm 1** IMPROVISED$^E$

---

**Definitions:**
- $b$: common public belief of player $P_1$ and player $P_2$
- $\mathcal{A}_i$: action space of $P_i$
- $s_i$: information state of $P_i$
- $b(s_1)$: belief of $P_2$ given $P_1$'s information state $s_1$
- $\pi$: joint blueprint policy
- $R(s_1, s_2, \pi, [a_1, a_2])$: reset current game state with $s_1$, $s_2$, rollout until termination following (the optional $[a_1, a_2]$ and then) $\pi$, and return the total reward.

**Method:**
initialize $q_\pi(a_1, a_2, b) = 0$ for $(a_1, a_2) \in \mathcal{A}_1 \times \mathcal{A}_2$
sample $M$ private state for $P_1$, $s_1^{(1)}, \ldots, s_1^{(M)} \sim b$
$P_\pi(a_1) = \frac{1}{M} \sum_{i=1}^{M} \pi(a_1 | b(s_1^{(i)}))$ for $a_1 \in A_1$
**for** $s_1^{(i)} \in s_1^{(1)}, \ldots, s_1^{(M)}$ **do**
    sample $N$ private state for $P_2$, $s_2^{(1)}, \ldots, s_2^{(N)} \sim b(s_1^{(i)})$
    $q_\pi(b, s_1^{(i)}) = \frac{1}{N} \sum_j R(s_1^{(i)}, s_2^{(j)}, \pi)$
    **for** $(a_1, a_2) \in \mathcal{A}_1 \times \mathcal{A}_2$ **do**
      **if** $P_\pi(a_1) \geq \epsilon_p$ **then**
        $q_\pi(a_1, a_2, b, s_1^{(i)}) = -\infty$
      **else**
        $q_\pi(a_1, a_2, b, s_1^{(i)}) = \frac{1}{N} \sum_j R(s_1^{(i)}, s_2^{(j)}, \pi, a_1, a_2)$
      **end if**
    **end for**
**end for**
**for** $(a_1, a_2) \in \mathcal{A}_1 \times \mathcal{A}_2$ **do**
    $q_\pi(a_1, a_2, b) = \frac{1}{M} \sum_i \max \left[ q_\pi(a_1, a_2, b, s_1^{(i)}), q_\pi(b, s_1^{(i)}) \right]$
**end for**
**for** $a_1 \in A_1$ **do**
    $f(b, a_1) = \text{softmax}_{a_2} \left[ q_\pi(a_1, a_2, b)/t \right]$
    $q_\pi(b, s_1, a_1) = \mathbb{E}_{s_2' \sim b(s_1), a_2 \sim f(b, a_1)} R(s_1, s_2', \pi, a_1, a_2)$
**end for**
**if** $\max q_\pi(b, s_1, a_1) \geq q_\pi(b, s_1) + \epsilon_q$ **then**
    **return** $\text{argmax}_{a_1} q_\pi(b, s_1, a_1)$
**else**
    **return** $a_1^{bp}$ // the action under blueprint
**end if**

---

# B  Experimental Details for Tiger-Trampoline

| Hyper-parameter | Values |
|---|---|
| learning rate | 0.0005, 0.0001 |
| batch size | 16, 32 |
| $\varepsilon$ annealing period | 20000, 10000 |
| RNN hidden dimension | 64, 32, 16 |

Table 2: Hyper-parameters of QMIX in the Tiger-Trampoline Experiment

In Section 5.1, we show the results of MAPPO and QMIX on the Tiger-Trampoline game. For the MAPPO we use the default parameters from the open sourced implementation[4] used for Hanabi,

---

[4]https://github. com/marlbenchmark/on-policy

except with a hidden size of 128, reducing the episode length cap, and reducing the number of threads by a factor of 2. For QMIX, we use the open sourced implementation[5] of the algorithm provided as part of the PyMARL framework [13]. We used the default agent and training configuration, except for the four hyper-parameters listed in table 2. For those, we tried all combinations of the corresponding values, producing a total of 24 runs, each training for 500k steps, or 250k episodes.

## C   Experimental Details for Finesse in Hanabi

In the Hanabi experiments, we implement IMPROVISED as follows (better viewed together with the pseudocode). The belief $b$ is the common public belief shared by *player 1* and *player 2* based on common knowledge available to all players and their common private knowledge of *player 3*'s hand. We first draw $M$ Player 2 hands $s_1'$ from $b$ and compute blueprint actions $a_\pi = \pi(b(s_1'))$ and $P_\pi(a)$. We then consider joint actions $\mathcal{A}_1 \times \mathcal{A}_2 = \{(a_1, a_2) | P_\pi(a_1) \leq 0\}$ for *player 1* and *player 2*. Since our goal is to find finesse style joint deviations, we further restrict $a_1$ to be a *hint move* to *player 3* and $a_2$ to be a *play move*. Given $s_1'$, *player 1* can further induce the private belief $b(s_1')$ over their own hand. For each of $s_1'$, *player 1* calculates Monte Carlo estimations of $q(a_1, a_2, b, s_1', )$ for $(a_1, a_2) \in \mathcal{A}_1 \times \mathcal{A}_2$ and $q_\pi(b, s_1')$ with $N$ samples drawn from $b(s_1')$. So far we have collected all the quantities required to compute the mapping $f$ for IMPROVISED$^P$ and for IMPROVISED$^E$. Finally, we draw another $K$ samples from the true $b(s_1)$ where $s_1$ now is the real hand of *player 2* to estimate $\delta = \max_{a_1} \mathbb{E}_{a_2 \sim f(b, a_1)} q_\pi(b, s_1, a_1) - q_\pi(b, s_1)$. *Player 1* will deviate to $\operatorname{argmax}_{a_1} \mathbb{E}_{a_2 \sim a_2^*(a_1)} q_\pi(b, s_1, a_1, a_2)$ if $\delta \geq 0.05$. In the next turn, *player 2* can carry out the same computation process to get $P_\pi(a_1)$ and $f(b, a_1)$ to figure out whether *player 1* has deviated and if so what is the correct response. *Player 1* and *player 2* do not share the random seed beforehand.

In the experiments where we run IMPROVISED on finesse-complete situations only, we set $M = 1000$, $N = 100$ and $K = 10000/|\mathcal{A}_1|$. It takes roughly 2 hours in total for both *player 1* and *player 2* to compute the deviations independently using 5 CPU cores and 1 GPU.

In the experiments where we run IMPROVISED on the full game of Hanabi, we reduce $M$ to 400 and share the result of $f(b, a_1)$ between Player 1 and Player 2 instead of computing it twice independently as we empirically find that the statistic is stable enough against random seeds. A full game then takes around 10-12 hours using 20 CPU cores and 2 GPUs.

## D   Societal Impact

We do not anticipate any immediate negative impact from this work.

---

[5]https://github.com/oxwhirl/pymarl