# OpenReview forum: "Self-Explaining Deviations for Coordination"
_NeurIPS.cc/2022/Conference — NeurIPS 2022 Accept_

### Official Review · Reviewer_tWnN · 2022-07-06

**Rating:** 8
**Confidence:** 4
**Soundness:** 4 excellent
**Presentation:** 3 good
**Contribution:** 4 excellent

**Summary:**

## Summary:

In multi-agent, partially-observable context, human beings are able to coordinate using very few information at their disposal, e.g. "minimal explicit agreement" and conventions, and reason about new information to extend previous conventions, allowing us to perform so-called *finesse* plays in the card game Hanabi.

This paper proposes a formal definition for such behaviours under the name **Self-Explaining Deviations (SED)**, within the formalism of Factored Observation Stochastic Games (FOSGs),  after applying a public POMDP transformation "that maps cooperative multi-agent settings to equivalent single-agent POMDPs".

Two assumptions are made though:
1. Sequential and publicly observable actions.
2. Only considers SEDs where the first player's private information is not necessary for coordination.

In short, under these assumptions, SEDs take place when, despite all players expecting a common knowledge blueprint (BP) policy (possibly a Nash equilibrium), an initial player's move deviates from the BP (and possibly from optimality), but it actually provides meaningful information that can be exploited by the subsequent players to choose moves that improves locally the performance, achieving a greater payoff.

The paper highlights the mechanics of SEDs thus defined on a toy example and proceeds to devise an optimization problem whose solving is tantamount to performing SEDs.
An approximated search algorithm is  thus devised to solve the optimization problem in a tractable fashion, at every player's turn, it is entitled IMPROVISED.

Two variants can be formulated though, IMPROVISED$^E$  and IMPROVISED$^P$, the former is based on the both players choosing there moves by maximizing the expected return in comparison to the BP's one, while the latter enforces the subsequent player to choose its move by maximising the expected improvement over the BP's action.

Note that the proposed algorithm relies on a blueprint policy being available, as it is a search algorithm that is added in a plug-and-play fashion after deployment.

Finally, the paper investigates the proposed algorithms ability to perform a subset of *finesse* plays in Hanabi.
Note that not all SEDs are *finesse* plays.
Comparisons are made against:
1. a Behaviour Cloning policy trained on human moves, and
2. SPARTA, another search algorithm applied to Dec-POMDP.
The BP policy used for that evaluation is the Behaviour Cloning policy.

The evaluation is performed in terms of number of specific *finesse* plays performed over a set of specific *finesse*-able situations.

Results show that the proposed methods largely outperform both methods in terms the number of *finesse* plays actually performed in *finesse*-able situations.

Also, using this proposed *finesse* benchmark, the paper investigates state-of-the-art MARL agents' ability to perform those specific *finesse* plays.
While all methods reach the same amount of *finesse*-able situations (~1.2k situations over 1k games), SAD, Other-Play, and OBL are only completing 43%, 33%, and 32% respectively. For comparison, the Behaviour Cloning blueprint reaches 85%.

## Post-Rebuttal and Initial AC-Reviewer Discussion Update:

Following the author's rebuttal, the author-reviewer discussion phases, and the AC-reviewer initial discussion phase, I mean to emphasise the following and detail how I have increased my marks:

1. Following the rebuttal, I acknowledged that the mathematical soundness/clarity issue that I was raising is **very subjective**, i.e. more a matter of taste than a matter of quality.
Indeed, the maths laid out in the paper are of a fairly good quality, I just would have hoped for better, but I recognised that it is not necessary given that the aim is primarily to communicate efficiently the concept that are presented in the paper, which I find they do perfectly.

2. The paper does not claim reaching a new SotA result on the Hanabi benchmark, yet my statistical significance concerns were regarding the results on the full game experiment benchmark, which I find **misguided**, in the end.

3. Rather, the paper does claim initialising a SotA result on the ability to perform finesse plays on the Hanabi benchmark, and the results on that end are statistically significant, without needing to run any test, given that they show a **5-fold increase** compared to the relevant baseline of SPARTA (and the sanity-check baseline of using solely the Behavior Cloning blueprint).

Thus, in light of this, I think that I mean to champion this paper for providing an excellent theoretical contribution (in formally framing what finesse plays are) and a statistically significant numerical evidence (in showing that their proposed algorithm raises the SotA in finesse-execution testing).

I am therefore updating the marks in my review:

* raising the mark from 6 to 8,

* raising the contribution mark from 3 to 4,

* raising the presentation mark from from 2 to 3.

* raising the soundness mark from 2 to 4.


**Questions:**

Please see section above for the major questions that are denoted (QX) with X being a digit.

**Limitations:**

I believe the authors have mainly adequately addressed the limitations and potential negative societal impact.
Nevertheless, I would welcome the paper to discuss further the possible limitations of the design, such as (i) the reliance on the shared function $f$ and the fact that it is chosen by the user, or (ii) the reliance on a blueprint policy (e.g. does it have to be the same for all players, and to what extend?), please?


**Strengths And Weaknesses:**

# Originality :
To my knowledge, the paper is the first to (i) formalise the in-famous *finesse* plays, (ii) propose an evaluation method for them in the game of Hanabi, (iii) propose to build an algorithm to maximise the likelihood of those *finesse* plays.

# Quality:
The quality of the paper is fair.
The math seem marginally sound on the surface, but I will highlight a clarity/soundness issues below.

Reproducibility is maybe a bit lacking due to the lack of details on the limitations and dependences of the algorithm proposed (e.g. the function $f$).

I find a strong issue with the strength of the numerical evidence:

--> Line 361 : Significance : In order to assert the actual significance of these results (despite the sample size being slightly smaller than in previous papers, e.g. 10k games in Other-Play,  cf. Figure 4), (Q1) could you perform statistical analysis of each distributions using, e.g. Kolmogorov-Smirnov tests [1], please ?
This would strongly strengthen my appreciation of the work and contribution here, and I would update my mark accordingly.

Similarly, considering the comparison with SPARTA as a baseline, I think it would be important to provide rollouts and time complexity statistics, please? In order to truly assert whether your proposed algorithms win on most norms or whether there are trade-offs to consider...

[1] : SciPy's two-sample Kolmogorove-Smirnov test : https://docs.scipy.org/doc/scipy/reference/generated/scipy.stats.ks_2samp.html

# Clarity & Soundness:
I have found the paper to be mainly well-written and to fairly substantiates its claims, but there are a few minor and major issues I would hope to see addressed:

--> line 60: need to specify $N \in \mathbb{N}$ as the number of players/agents

## --> line 153 : Soundness & Clarity : Timestep in FOSG VS in single-agent public POMDP :

Given the public POMDP transformation, aren't the players playing simultaneously for the joint decision rule to be defined?

I fail to understand what information are the joint decision rule actually considering.
Indeed, the notation defined around line 84 highlights that the joint decision rule is only using as input the set of information state sets over all players, but the equations (4) and (6) that defines how Bob plays highlights a dependence on Alice's action $a_1$ which is not part of any information state set yet, since it corresponds to a constituant of the next single-agent public POMDP timestep, no?

Or, alternatively, could you clarify when are the information state sets being updated?

I think it brings us back to how the history is being defined as its reliance on the time step $t$ , in the FOSG, makes it ambiguous between each player's turn, if they play sequentially ?
i.e. Alice plays at time step $t$ and Bob at time step $t+1$ in the FOSG, but the joint decision rule in the equivalent single-agent public POMDP  lacks clarification of when are the information state set computed, maybe?

(Q2) If you understand the soundness and clarity issue that I raise here, then may I propose the following:
1. For clarification, it mind be better to state the Assumption 1 with respect to the FOSG (and maybe place it in the text before introducing the public POMDP transformation).
2. For soundness, it mind be better to rewrite the joint decision rule as $a \equiv \Gamma(s_1^{t}, s_2^{t+1}, ... , s_N^{t+N})$

## --> line 177 : Clarity : Function $f$ :
(Q3) Could you clarify how is $f$ computed? Is it shared between players?

I remember line 173 stating that it is 'determined by the algorithm designer', but I am not sure I understand the meaning: does it mean that it is like an hyperparameter that needs to be provided for the algorithm to work?
If so, I would flag it as a critical limitation that needs to be discussed and better detailed, please.
I would welcome a very detailed table maybe showing the mapping that this function performs, if it is relevant?


## --> Equation (1) : Clarity & Soundness : Operator $\circ$
I think that leaving the $\circ$ operator undefined is hurting the clarity and soundness of the claim.

Indeed, its usage on $\Gamma_1$ is ambiguous as $\Gamma_1$ outputs on action $a\in\mathcal{A}$, which is not from the input space of $\Gamma_2$.
I think this boils down to the lack of rigour when stating the dependencies of the history on the timestep $t$ (in the FOSG), e.g.--> line 67 : rendering the the history variable as  $h_t$, given the dependence on the time step $t$ in its constituants. And therefore, the information state set would need such an addition too...

## --> line 234-237 : Clarity : Optimality Argument lacking substance
I am afraid I do not understand the optimality argument that is made in this paragraph. Could you clarify further please, as it is fairly central to the contribution, right?
(Q4) Could a proof of the optimality be in order, or maybe just the use of some formalism to ground the text to the equations discussed so far, please?


## --> line 244-245 : Clarity : Jensen Inequality & Alice's conditional behaviour
I am afraid I do not understand the causality link made by this piece of text, nor how Jensen's inequality illustrates Alice's conditional behaviour. It might be obvious but I unfortunately cannot see it in the present form, thus (Q5) may I request a clarification, please?

--> line 245: clarity : shouldn't it be an argmax instead of max?

## --> Figure 3: Result Transparency :
I would suggest the author to display their algorithms' results in Figure 3 (as well as the training curve of the blueprint Behavior cloning alone as a baseline, maybe?), in order to be complete and transparent.

## --> Table 1.b : Clarity and Soundness : Major Issue/Misunderstanding
I fail to make sense of the Blueprint policy suddenly performing 0 finesse out of finesse-able situations, while it is shown to perform 85% of those as presented in Table 1.a.

(Q6) Or could it be that I am making a confusion between "situations where a finesse play is performed" and finesse-complete situations, don't they mean the same?
Could it be that they are different depending on which player's action is being evaluated, i.e. the player that initiate the finesse play and peform an SED being opposed to the player(s) who recognises the SED and acts accordingly.

In any case, I would recommend the authors to try to better organise (with subtitles or bold summarising statements at the start of paragraphs, maybe?) the Section 5.4 ; it was difficult to read and make sense of, I must confess :S.
Maybe all the information are too densely packed in, it might need more breathing room for some examples, e.g. an example of the typical play being evaluated with a turn-by-turn analysis.

Also, the more I read the paper, the more I am pondering whether evaluation for finesse (in the general sense) should be addressing each players' turn separately?
Indeed, I am assuming it is one thing (a) to realise that a finesse (coordination) play can be performed in a given situation and then be the one to initiate it (player 1), but it is an entirely other thing (b) to be able to understand the resulting SED that occured and choose the right move to maximise payoff (player 2), no?

I am still feeling confused by the function $f$, which is involved in (b) for more than half of the task. I feel the need for it to be better detailed, in that regard too, please?


# Significance :
Most of the results provided may be more compelling if the sample size was similar to previous papers.
That being said, the theoretical significance of the paper is high, but I would really like to see the significance of its results be better asserted.
Please consider addressing the following points:

## --> Line 353 : Missing Results for Human-Cooperation Significance :
What about IMPROVISED$^P$, since it is more valuable for cooperation with humans?

Could you report the number and percentages of finesse-able / finesse-complete situations encountered during these 100 games, please? like in Table 1.a

I think that it would be important to perform this evaluation on a greater number of games, similarly to previous works, in order to assert the statistical significance of the results.

## --> Line 356 : Selfplay vs Crossplay & Zero-Shot Coordination :
As only selfplay scores are presented, it is unclear whether the blueprint agents have developed idiosyncratic policies or not (especially if the behavioral cloning training data only contains one pair of players, for instance ?).

Thus, to provide more transparency and be more in touch with the MARL field's dilemma at large (of which zero-shot coordination is a major problem, especially on the Hanabi benchmark), could you present crossplay performance results (with differently seeded blueprint agents), like in the Other-Play paper, for instance, please?
Unless maybe it is irrelevant as the algorithm your propose manually computes its own belief at every move and the Monte Carlo rollouts are robust to noise, or something of the kind?

I understand that this is going the extra step, and I would not mind if the authors were not able to address it...

## --> Line 361 : Significance : Statistical Significance Tests :
see section Quality above...

---

> ### Author Response · Authors · 2022-08-02
> **Response to Reviewer tWnN (1/2)**
>
> Thanks for your extremely thorough review! We really appreciate the time and effort you put into giving us this feedback!
>
> > Given the public POMDP transformation, aren't the players playing simultaneously for the joint decision rule to be defined?
>
> Technically, yes. But in settings with sequential moves, this amounts to the other players having only a single legal action that “no-ops”. It is possible to construct a public POMDP in which the players act sequentially. These issues are mostly notational. We added a footnote about this for our revision.
>
> > the equations (4) and (6) that defines how Bob plays highlights a dependence on Alice's action  which is not part of any information state set yet, since it corresponds to a constituent of the next single-agent public POMDP timestep, no?
>
> True, but, if Alice plays that action, it will be part of Bob’s information state at the *next* timestep. In other words, the agents are coming up with a plan for both the current decision and the next decision in the game. Similarly, in chess, for example, one could plan about decision points that condition on actions that haven’t happened yet.
>
> > Or, alternatively, could you clarify when are the information state sets being updated?
>
> Each player’s information state is updated once per time-step. However, this does not preclude the agents from doing forward planning.
>
> > I think it brings us back to how the history is being defined as its reliance on the time step  , in the FOSG, makes it ambiguous between each player's turn, if they play sequentially ? i.e. Alice plays at time step  and Bob at time step  in the FOSG, but the joint decision rule in the equivalent single-agent public POMDP lacks clarification of when are the information state set computed, maybe?
>
> They play *sequentially*. We added a sentence to clarify this into our revision.
>
> Please let us know if our clarification here is insufficient.
>
> > (Q3) Could you clarify how is f computed? Is it shared between players? ..  I remember line 173 stating that it is 'determined by the algorithm designer', but I am not sure I understand the meaning: does it mean that it is like an hyperparameter that needs to be provided for the algorithm to work?
>
> In line 170, f is the function that defines Bob’s response to Alice’s deviation. At this point in the submission, we are not yet committing to a particular algorithm. Rather, we are formalizing the structure that a self-explaining deviation needs to satisfy. The main point here is emphasizing that Bob’s response conditions on the public belief state and Alice’s action. Later on in the submission, we get into the specifics of how f is structured for IMPROVISED, the algorithm that we propose. For IMPROVISED, f is fully specified.
>
> > I think that leaving the \circ operator undefined is hurting the clarity and soundness of the claim. Indeed, its usage on \Gamma_1  is ambiguous as \Gamma_1 outputs on action a \in \mathcal{A}, which is not from the input space of \Gamma_2.
>
> We use \circ to notate function composition. We added a sentence to clarify this to our revision.
> To clarify, as stated in the above comments, \Gamma_1 here is the prescription for time t, while \Gamma_2 is the prescription for time t+1.
> As to the second point, in general, in the public POMDP, \Gamma_2 may condition on more information than just Alice’s action. However, due to our assumptions on the structure of Bob’s response (i.e., line 170), \Gamma_2 uses only Alice’s action to respond.
>
> > I am afraid I do not understand the optimality argument that is made in this paragraph. Could you clarify further please, as it is fairly central to the contribution, right?
>
> Yes, thanks for pointing out that this was unclear! In sections 4.2 and 4.3, we are discussing a case in which Alice’s deviation is defined by a single action. Optimal is a confusing term for us to have used here. We removed this to avoid confusion.
>
> > I am afraid I do not understand the causality link made by this piece of text, nor how Jensen's inequality illustrates Alice's conditional behaviour. It might be obvious but I unfortunately cannot see it in the present form, thus (Q5) may I request a clarification, please?
>
> Of course! Jensen’s inequality says that f(E[X]) <= E[f(X)], where E is expectation, f is a convex function and X is a random variable. In our context, the maximization over a_1 is a convex function, so moving it inside of the expectation cannot decrease the value of the expression. In practice, this translates to saying: if Alice is allowed to observe her own information state before selecting the best action, she can potentially achieve a higher expected return than if she were forced to select the best action in a fashion agnostic to her information state.

---

> > ### Author Response · Authors · 2022-08-02
> > **Response to tWnN (2/2)**
> >
> > > I fail to make sense of the Blueprint policy suddenly performing 0 finesse out of finesse-able situations, while it is shown to perform 85% of those as presented in Table 1.a. .. (Q6) Or could it be that I am making a confusion between "situations where a finesse play is performed" and finesse-complete situations, don't they mean the same? Could it be that they are different depending on which player's action is being evaluated, i.e. the player that initiate the finesse play and peform an SED being opposed to the player(s) who recognises the SED and acts accordingly.
> >
> > Table 1a shows the number of finessable situations that were reached – not the number of finesseses that actually occurred. Importantly, it is not possible to perform a finesse on every turn in Hanabi. It is only in specific situations that finesses (or more generally SEDs) are possible. It is the number of these specific situations that is being shown in Table 1a.  In contrast, table 1b shows the proportion of the time that a finesse occurred, conditioned on being in a finessable situation.
> >
> > > In any case, I would recommend the authors to try to better organise (with subtitles or bold summarising statements at the start of paragraphs, maybe?) the Section 5.4 ; it was difficult to read and make sense of, I must confess :S. Maybe all the information are too densely packed in, it might need more breathing room for some examples, e.g. an example of the typical play being evaluated with a turn-by-turn analysis.
> >
> > Thanks for this feedback! We have added headers for our revision.
> >
> > > Indeed, I am assuming it is one thing (a) to realise that a finesse (coordination) play can be performed in a given situation and then be the one to initiate it (player 1), but it is an entirely other thing (b) to be able to understand the resulting SED that occured and choose the right move to maximise payoff (player 2), no?
> >
> > Indeed! They are two different things. We are only counting finesses if *both* players complete their parts.
> >
> > > Line  353 What about IMPROVISED^P, since it is more valuable for cooperation with humans? Could you report the number and percentages of finesse-able / finesse-complete situations encountered during these 100 games, please? like in Table 1.a. I think that it would be important to perform this evaluation on a greater number of games, similarly to previous works, in order to assert the statistical significance of the results.
> >
> > We weren’t able to run both IMPROVISED versions on full games due to its computational cost. We updated the paper with additional seeds. Now Table 1(b) is evaluated on 1000 seeds, and we find that IMPROVISED^P is not noticeably better than IMPROVISED^E for performing SEDs. We have updated the related discussions in the paper. When running IMPROVISED^P on the entire game, it performs 21 finesse out of 100 games. IMPROVISED is quite expensive to run on full games and the main purpose of this paper is to define and propose the first algorithm for SED. More future work could be done to design faster algorithms that can be more easily applied on full games.
> >
> > > Thus, to provide more transparency and be more in touch with the MARL field's dilemma at large (of which zero-shot coordination is a major problem, especially on the Hanabi benchmark), could you present crossplay performance results (with differently seeded blueprint agents), like in the Other-Play paper, for instance, please? Unless maybe it is irrelevant as the algorithm you propose manually computes its own belief at every move and the Monte Carlo rollouts are robust to noise, or something of the kind?
> >
> > This is a really great idea for an experiment! We would also love to have an experiment like this but cannot add it due the limit of time. Our experiments mostly use a behavior clone agent trained from human data and from our experience the supervised learned policies are quite consistent across different training seeds. Therefore the zero-shot coordination may be less of an issue in our current setting. However, we believe that SEDs should definitely be studied in the zero-shot coordination setting in the future when the blueprint agents are learned from scratch without human data.
> >
> > > Line 361 : Significance : Statistical Significance Tests :
> >
> > We are able to run additional seeds for Table 1(b) and Line 361.  Both results are updated with 1000 seeds. IMPROVISED^E and IMPROVISED^P achieve average scores of 18.08 +/- 0.28 and  18.18 +/- 0.27 respectively while the blueprint gets 17.18 +/- 0.28 on those same games. Number after +/- is standard error. A paired t-test of mean(diff) - 1.96 * std(diff) / sqrt(n) returns 0.14 for IMPROVISED^E vs BP and 0.25 for IMPROVISED^P vs BP.

---

> > > ### Comment · Reviewer_tWnN · 2022-08-09
> > > **Response to Authors**
> > >
> > > Thank you for your thorough and detailed responses to my questions and comments.
> > > I address some subsequent points below:
> > >
> > > ### Mathematical Soundness/Clarity:
> > >
> > > I appreciate the authors revision to the paper and the answers made to my questions, but I still find, albeit very subjectively, that the revised paper does *not sufficiently* address my soundness/clarity issue raised around (Q2) in my initial review.
> > >
> > > I mean to emphasise that it is probably a matter of taste between laying out maths that convey the relevant meaning (for communication purpose only), or laying out maths that are undisputable and rigorous (for theoretical proving purpose).
> > > Therefore, as the formalism in the paper is mainly there for communication purpose, I fully understand and I am tempted to agree with the authors choice, especially given the space limitations.
> > >
> > >
> > > ### Computational Cost :
> > >
> > > In appendices, you provided some details about the computational cost of the algorithm, but I find them difficult to make sense of in an absolute fashion, thus I would recommend trying to compare them with SPARTA numbers, maybe, for instance, or another baseline.
> > >
> > > ### Crossplay Experiment Request:
> > > I appreciate the authors response and agree that this request is not accessible, nor necessary in the end, given the far too great computational cost of the approach (it will be more valuable for subsequent iterations of the algorithm, once tractability issues are fixed).
> > >
> > > ### Statistical Significance Test :
> > > I appreciate the authors efforts to increase the number of seeds and the number of games, despite the computational cost of the method.
> > > I also appreciate the effort in running paired t-test, but I do not find a paired t-test to be convincing in this context, as it is only able to compare the mean of two distributions, which is far less informative than being able to compare two distributions as a whole (e.g. bimodal vs unimodal with similar means would result in a similarity result under a paire t-test, whereas the two distributions are clearly not similar), which is what a two-sample KS test can do, as far as I understand.
> > >
> > > Thus, I appreciate the authors revisions to the paper, but my main concern of quality has not been addressed fully, as of now.
> > > I assume that it can easily be done for a possible camera-ready version though.
> > > Whilst I cannot raise my rating of the paper as is, I mean to emphasise for subsequent review steps that I am very much willing to increase my rating from 6 to 7, provided the final version contains the requested tests...

---

### Official Review · Reviewer_y99U · 2022-07-12

**Rating:** 6
**Confidence:** 3
**Soundness:** 3 good
**Presentation:** 3 good
**Contribution:** 3 good

**Summary:**

This paper introduces the idea of a “self-explaining deviation” [SED] where an agent does an action that is seemingly irrational with the explicit intention of getting other agents to realize that the (privately observed) state is not what the others think it is, and make them react accordingly. The authors provide a fairly generic definition of an SED, and an algorithm for recognizing such situations and altering the policy accordingly. They provide an evaluation of their approach in both a simple toy problem and the game of Hanabi where this kind of action is common among human players, and not produced by previous computational approaches.


**Questions:**

The evaluation is conducted in environments were you have an extremely precise model of the environment; would any of this work (even the probabilistic IMPROVISED^P) where the environment model is only learned (or, how good does the model have to be before any of this is applicable?)


**Limitations:**

Authors address this directly, no issues.

**Strengths And Weaknesses:**

Originality
I’m not aware of work conceptualizing anything similar to SED in either POMDP-adjacent systems, or earlier approaches to recursive modeling of the actions of other agents.

Clarity
In general the paper is written clearly, with good examples. Most of the mathematical content is also fine, although I find4.3/4.4 the most confusing part of the paper, with few examples and given that the rest is fairly straightforward. Presumably the appendix information makes this more clear.

Quality
I don’t find any obvious technical errors in the main body of the paper. There could be a greater discussion in the main body of the computational issues of the two versions of the IMPROVISED algorithm and how they would relate to general scalability of the approach. (this is pointed out in the conclusions, but nothing appears in the main body).

Significance
This is probably the biggest issue? Yes, the authors give a horrifying realistic (but extremely specific) human example, but otherwise confine the analysis to the relatively cerebral and abstract Hanabi. There is a bit of an odd focus in the current DecPOMDP/MARL literature in finding some environment that eliminates or limits explicit communication even though many practical applications don’t have that as a limitation.

---

> ### Author Response · Authors · 2022-08-02
> **Response to y99U**
>
> Thanks for your thoughtful comments! We respond below.
>
> > There could be a greater discussion in the main body of the computational issues of the two versions of the IMPROVISED algorithm and how they would relate to general scalability of the approach.
>
> We include the computational cost of the IMPROVISED in the appendix. The two versions cost roughly the same amount of compute. In its current Monte Carlo rollout based form, they are much more expensive than single agent SPARTA but should be comparable to other joint search algorithms such as multi-agent sparta. We noted the high computational cost in the limitation and future work section.
>
> > There is a bit of an odd focus in the current DecPOMDP/MARL literature in finding some environment that eliminates or limits explicit communication even though many practical applications don’t have that as a limitation.
>
> We agree that explicit communication is oftentimes a more relevant setting to real world applications. However, if an explicit communication channel is available, there really is no need for SEDs, as the acting agent can simply state how the situation is abnormal. Counterexamples are settings like the Pizza example, where a channel is available but can’t be used trivially due to the presence of an adversary.
>
> > The evaluation is conducted in environments were you have an extremely precise model of the environment; would any of this work (even the probabilistic IMPROVISED^P) where the environment model is only learned (or, how good does the model have to be before any of this is applicable?)’
>
> We believe that all of this would work in a setting where the model of the environment is learned. Of course, as is also the case with single-agent reinforcement learning, the effectiveness of the method would certainly be dependent on the quality of the model. It is difficult to say in general exactly how accurate a learned model would need to be for the approach to be effective, as is generally the case with model-based RL approaches.

---

### Official Review · Reviewer_4pB2 · 2022-07-14

**Rating:** 6
**Confidence:** 3
**Soundness:** 3 good
**Presentation:** 3 good
**Contribution:** 4 excellent

**Summary:**

This paper focuses on a specific subclass of coordination problem -- self-explaining deviations (SEDs). SEDs refer to the actions that deviate from reasonable behavior but with the intention of communicating with other agents. This paper first motivates SED with a vividly real-world example and afterward gives a formal definition of it. Then they proposed a novel algorithm to perform SEDs. Their further evaluations of toy games and Hanabi show the strengths of the proposed algorithms.

**Questions:**

Some questions are listed above, and here are other questions.

1. Is it possible to extend the idea of SEDs to more than 2 players in Definition 1?
2. Is it a common assumption that 'Bob can detect the deviation as $\mathcal{D}(b)$' in Line 211?
3. What do the dash lines between Bob in Figure 2 mean?
4. Can other search algorithms perfectly solve Trampoline Tiger?

**Limitations:**

The authors have described enough limitations of their algorithms. And completely solving these limitations might be enough for a new paper.

**Strengths And Weaknesses:**

Strengths

1. **Importance of the problem**. The problem this paper considers is important. Humans have the ability to coordinate with others using minimal explicit agreement and only relying on information from normal or abnormal actions. This paper formalizes the definition of such behavior as self-explaining deviations and steps forward in this interesting topic.
2. **Motivation.** This paper is well-motivated. SEDs are very common behaviors of humankind but are rare for AI. Explicitly incorporating SEDs into other baselines might further improve performance.
3. **The novelty of the proposed method**. The proposed method is simple but novel.
4. **Clarity**. This paper is generally well-written and easy to follow.



Weaknesses

1. **The soundness of the proposed method**. The reviewer thinks the assumptions might be too restrictive, in particular, Assumption 2. What if we remove Assumption 2? How likely is it that the game in real life meets this assumption?
2. **Quality and soundness of empirical evaluation**. (1) The provided results are sound, but the evaluations are conducted only on two games, including a toy game. Can the author show more results of other games or more interpretations of existing results? It is okay if the rebuttal time is not enough for the authors to do extra evaluations. (2) It seems that the algorithms and evaluations are based on two cooperative players. Is it possible to extend this idea to more players?

---

> ### Author Response · Authors · 2022-08-02
> **Response to 4pB2**
>
> Thanks for your thoughtful comments! We respond below.
>
> > The reviewer thinks the assumptions might be too restrictive, in particular, Assumption 2. What if we remove Assumption 2? How likely is it that the game in real life meets this assumption?
>
> Assumption 2 acts as a regularizer on the complexity of the deviation. We suspect that most real life SEDs are probably relatively low complexity, as complex SEDs run higher risks of miscoordination. E.g., the real life example in Figure satisfies assumption 2, as the deviation does not involve different responses based on the dispatcher’s private information.
>
> > Can the author show more results of other games or more interpretations of existing results? It is okay if the rebuttal time is not enough for the authors to do extra evaluations.
>
> We can add more interpretations (see comments to reviewer 1). Investigating other games is trickier because of the lack of suitable MARL benchmarks for SEDs. For a start fully cooperative, partially observable benchmarks that are coordination problems are rare. The community has largely relied on the SMAC benchmark which does not pose coordination challenges as defined in our work and has been shown to have very limited partial observability (e.g. feed-forward networks or open-loop policies do well in many maps).
>
> > It seems that the algorithms and evaluations are based on two cooperative players. Is it possible to extend this idea to more players? … Is it possible to extend the idea of SEDs to more than 2 players in Definition 1?
>
> The algorithm is trivially extendable to a setting with an arbitrary number of players, so long as there are only two players participating in the SED. Extending the algorithm to settings in which more than two players participate in the SED is also possible, but would become more expensive, as it would require reasoning over a larger number of possibilities. Investigating extensions that are scalable in the number of participating players is an interesting direction.
>
> > Is it a common assumption that 'Bob can detect the deviation as D(b)' in Line 211?
>
> That Bob can detect the deviation follows from Assumption 1 and Definition 1. In particular, Assumption 1 says that actions are publicly observable (so Bob observes Alice’s actions). Definition 1 says that an initiating SED action must have negligible probability under the blueprint policy, given the public belief state. Thus, Bob can both observe Alice’s action, and is aware of the fact that this action has negligible probability under “normal circumstances”.
>
> > What do the dash lines between Bob in Figure 2 mean?
>
> This is a convention for extensive-form games to indicate that Bob is unable to distinguish between the two trajectories (and thus must use a decision rule for both). We state this in the caption: “ Dotted lines between two Bob nodes means that Bob cannot distinguish them.”
>
> > Can other search algorithms perfectly solve Trampoline Tiger?
>
> Yes! For example, see  “Optimally solving dec-pomdps as continuous-state mdps”. Unfortunately, the class of search algorithms that can handle Trampoline Tiger-style problems is difficult to scale to large settings.

---

> > ### Comment · Reviewer_4pB2 · 2022-08-07
> > **Thanks for the detailed clarifications.**
> >
> > The responses have addressed most of my concerns.

---

### Official Review · Reviewer_t5d6 · 2022-07-14

**Rating:** 6
**Confidence:** 4
**Soundness:** 3 good
**Presentation:** 3 good
**Contribution:** 2 fair

**Summary:**

This paper describes the problem of discovering and acting on "self-explaining deviations" (SEDs) when two or more agents are cooperating in a turn-taking environment such that one agent can deviate from its expected behavior in order to signal to another agent that it has private information indicating that a higher utility situation can be achieved if the other agent also deviates from its expected behavior.  The factored observation stochastic game is modeled as a public POMDP to plan behaviors for all agents jointly.  Two flavors of an algorithm called IMPROVISED are provided to enable the signaling agent to discover such opportunities and enable the signaled agent to  know how to respond to achieve the higher utility.  The approach is evaluated in the context of (1) a novel Trampoline Tiger problem, demonstrating that even in simple scenarios, state-of-the-art deep MARL algorithms like MAPPO and QMIX cannot learn to find and exploit SEDs, and (2) the card game Hanabi, demonstrating that IMPROVISED can find SEDs called finesse plays commonly used by human players but not again not learned by deep MARL approaches and rarely even by a learned model that is trained to replicate human play from observed games.

**Questions:**

1) Have you analyzed any other SEDs discovered during Hanabi?  How might those SEDs be characterized, and do they follow human wisdom for the game, or have novel strategies been discovered?

2) Where there other types of deviations discovered by IMPROVISED in Hanabi that might not be SEDs?  Could the algorithm be useful for discovering and employing other deviations from blueprint strategies?

3) Have you conducted any statistical significance testing of your results (for sake of completeness of the analysis)?

4) In the Trampoline Tiger problem, is the general lack of discovery of the optimal strategy a problem of not balancing the exploration-exploitation tradeoff properly, or something else?  Given the small size of the game, I would have expected RL to eventually find the optimal strategy and then continue to exploit it.

**Limitations:**

The limitations of the approach are adequately described.

**Strengths And Weaknesses:**

Overall, the research is interesting and is relevant to the multiagent planning and reinforcement learning communities at NeurIPS.  The problem and solution are novel as far as I'm aware, and are also relevant to human-agent connections in AI.  The paper is relatively clear and easy to follow.

Given the lack of theoretical analysis, I would have maybe liked to see a little more evaluation on the empirical side.  I appreciated the use of two domains -- one toy problem for illustrating the challenge for state-of-the-art RL methods, as well as a more challenging game playable by humans.  But the paper generally lacked any statistical significance testing (I suspect you'll find improved performance in terms of using finesse, but it is less clear about the difference in overall game scores).  I would have especially appreciated more discussion of the other types of SEDs found in Hanabi and whether those are also categories of moves played by humans or if the algorithm discovered novel strategies that can push the boundary of human play (similar to innovations in backgammon and go by AI agents throughout the years).  Instead, the paper mentions "it can be either that IMPROVISED finds no beneficial deviations or it finds better, non-finesse SEDs".  I suspect the latter happened at least some of the time, given the overall boost in game score compared to the blueprint strategies.

Minor notes:

-- line 157: "where the Bob's" => "where Bob's"
-- line 366: "Dec-POMDs" => "Dec-POMDPs"

---

> ### Author Response · Authors · 2022-08-02
> **Response to Reviewer t5d6**
>
> Thanks for your thoughtful comments! We respond below.
>
> > I would have especially appreciated more discussion of the other types of SEDs found in Hanabi and whether those are also categories of moves played by humans or if the algorithm discovered novel strategies that can push the boundary of human play (similar to innovations in backgammon and go by AI agents throughout the years).
>
> IMPROVISED indeed finds many other types of SEDs that do not match the exact definition of finesse. For example, the IMPROVISED^P in Table2 deviates 142/200 (71%) and 62 out of those 142 (44%) deviations are finesses. Finesse refer to the exact situation where player 1 hints to player 3 about an unplayable card player 2 plays their **latest** card that has not been hinted. IMPROVISED performs other types of SEDs. For example, player 2 may play a non-latest card, such as 2nd or 3rd newest card, or the latest card may have been hinted at but player 2 would otherwise not have played it if player 1 had not deviated. Player 1 may also choose to hint at a different card that will not lead to a traditional finesse but still maximize the average return (in IMPROVISED^E) or average probability of improvement (IMPROVISED^P) upon player 2 executing their part of the deviation. Some of these SEDs are quite easy for humans to understand but others may seem novel to humans as they utilize the power of exact belief tracking and Monte Carlo rollouts that are difficult for humans to perform. Fully understanding the differences between human-like finesses and SEDs as defined in our work is a great direction for future work.
>
> > discussions on other types of SEDs
>
> IMPROVISED indeed finds many other types of SEDs that do not match the exact definition of finesse. For example, the IMPROVISED^P in Table2 deviates 142/200 (71%) and 62 out of those 142 (44%) deviations are finesses. Finesse refer to the exact situation where player 1 hints to player 3 about an unplayable card player 2 plays their **latest** card that has not been hinted. IMPROVISED performs other types of SEDs. For example, player 2 may play a non-latest card, such as 2nd or 3rd newest card, or the latest card may have been hinted at but player 2 would otherwise not have played it if player 1 had not deviated. Player 1 may also choose to hint at a different card that will not lead to a traditional finesse but still maximize the average return (in IMPROVISED^E) or average probability of improvement (IMPROVISED^P) upon player 2 executing their part of the deviation. Some of these SEDs are quite easy for humans to understand but others may seem novel to humans as they utilize the power of exact belief tracking and Monte Carlo rollouts that are difficult for humans to perform. Fully understanding the differences between human-like finesses and SEDs as defined in our work is a great direction for future work.
>
> > Were there other types of deviations discovered by IMPROVISED in Hanabi that might not be SEDs? Could the algorithm be useful for discovering and employing other deviations from blueprint strategies?
>
> The deviations discovered by IMPROVISED naturally satisfy the definition of SEDs by design.
>
> > Have you conducted any statistical significance testing of your results (for sake of completeness of the analysis)?
>
> We ran additional experiments and now the results in Table1(b) are updated with 1000 samples. The numerical results have also been updated to include standard errors. IMPROVISED^E and IMPROVISED^P achieve average scores of 18.08 +/- 0.28 and  18.18 +/- 0.27 respectively while the blueprint gets 17.18 +/- 0.28 on those same games. Number after +/- is standard error. A paired t-test of mean(diff) - 1.96 * std(diff) / sqrt(n) returns 0.14 for IMPROVISED^E vs BP and 0.25 for IMPROVISED^P vs BP.
>
> >In the Trampoline Tiger problem, is the general lack of discovery of the optimal strategy a problem of not balancing the exploration-exploitation tradeoff properly, or something else?
>
> Due to the reward landscape (certain actions have devastating consequences and negative rewards of those actions are much bigger than the positive rewards) and other multi-agent challengings such as  both agents exploring and evolving at the same time, RL almost certainly first find the suboptimal equilibrium of "Never jump, never pull lever". Then escaping that equilibrium requires both agents to jointly deviate in a very specific way. Whereas unilateral deviation (such as any form of epsilon greedy or other random exploration) is highly negative EV, so the RL algorithms are continually reinforced to stay stuck at the suboptimal equilibrium even if they explore.

---

> > ### Comment · Reviewer_t5d6 · 2022-08-08
> > **RE: Author's Response**
> >
> > I thank the authors for their responses to my questions and comments.
> >
> > Other SEDS: Thanks for the analysis.  Since the paper includes some qualitative analysis of the play of the agents using the algorithm, I think a little discussion about the types of SEDs found, especially the novel ones or those that differ from (good) human play, would strengthen the discussion in the paper and highlight the importance of using such a method.
> >
> > Statistical testing: I appreciate the additional analysis, which would also be good to add to the paper.  It appears that finding SEDs didn't quite improve performance (given that the p-values are above 0.1), but that wasn't the main goal of the work.
> >
> > Trampoline Tiger: I better understand now!

---

### Meta-Review · Area_Chair_4Gtu · 2022-08-26

**Recommendation:** Accept
**Confidence:** Less certain

**Metareview:**

The reviewers carefully analyzed this work and agreed that the topics investigated in this paper are important and relevant to the field. All reviewers generally shared a positive impression of this work. One reviewer mentioned, as possible limitations, the lack of theoretical analyses and more empirical evaluation. They also (initially) pointed out that the paper lacked statistical significance testing. After the authors' rebuttal, however, the reviewer said they better understood some aspects of the paper, that they appreciated the additional statistical testing analyses, and suggested adding more discussion on such points. Another reviewer agreed that this paper investigates an important problem and that it formalizes many relevant related concepts. As limitations, the reviewer said that the method requires assumptions that may be too restrictive (e.g., Assumption 2). They were also concerned since evaluations were conducted on only two games, one of which is a toy domain. Nevertheless, the reviewer was satisfied with the authors' responses. A third reviewer acknowledged that this work is novel, that the paper is well-written, and that its mathematical contents are well presented. After reading the authors' rebuttal, the reviewer was still concerned with the fact that the computational costs were discussed only for specific runs (and are a significant limitation, as the authors noted). This, the reviewer believes, suggests that the method might have somewhat limited practical applicability. A fourth reviewer brought up many technical questions as part of their review, which (at the time) they thought had not been adequately addressed in the authors' rebuttal. However, the reviewer later said that some of their criticisms regarding the formalization proposed in the paper were not a sufficient reason for rejection. They recognized that resolving such issues was not necessary given that this work aims primarily to communicate the main high-level ideas introduced by the authors. As final feedback to the authors, this reviewer suggested that the authors could provide an open-source implementation of the finesse-execution testing benchmark, so that subsequent works can build upon this new metric and evaluate their algorithms against it. Overall, thus, it is clear that most reviewers were positively impressed with the quality of this work and look forward to an updated version of the paper that addresses the suggestions mentioned in their reviews and during the discussion phase.

**Award:**

No

---

### Decision · Program_Chairs · 2022-09-14

Accept